# DATA INTERPOLATING PREDICTION: ALTERNATIVE INTERPRETATION OF MIXUP

**Takuya Shimada**\*
The University of Tokyo

**Shoichiro Yamaguchi**
Preferred Networks

**Kohei Hayashi**
Preferred Networks

**Sosuke Kobayashi**
Preferred Networks

## ABSTRACT

Data augmentation by mixing samples, such as Mixup, has widely been used typically for classification tasks. However, this strategy is not always effective due to the gap between augmented samples during training and clean samples. This gap may prevent a classifier from learning the optimal decision boundary and increases the generalization error. To overcome this problem, we propose an alternative framework called Data Interpolating Prediction (DIP). Unlike common data augmentations, we encapsulate the sample-mixing process in the hypothesis class of a classifier so that train and test samples are treated equally. We derive the generalization bound and show that DIP reduces the original Rademacher complexity. Also, we empirically demonstrate that DIP can outperform existing Mixup.

## 1 INTRODUCTION

Data augmentation (Simard et al., 1998) has played an important role in training deep neural networks for the purpose of preventing overfitting and improving the generalization performance. Recently, sample-mixed data augmentation (Zhang et al., 2018; Tokozume et al., 2018a;b; Verma et al., 2018; Guo et al., 2019; Inoue, 2018) has attracted attention, where we combine two samples linearly to generate augmented samples. The effectiveness of this approach is shown especially for image classification and sound recognition tasks.

Many traditional data augmentations (e.g., slight deformations for image data (Taylor & Nitschke, 2017)) rely on the specific properties of the target domain such as invariances to some transformations. On the other hand, sample-mixed augmentation can be applied to any dataset due to its simplicity. However, its effectiveness depends on the specified data structure. There is basically a difference between original clean samples and augmented samples in that augmented samples are not drawn from an underlying distribution directly. Thus a classifier trained with sample-mix augmentation may learn the biased decision boundary. In fact, we can easily create a distribution where sample-mix deteriorates the classification performance (See Fig. 1).

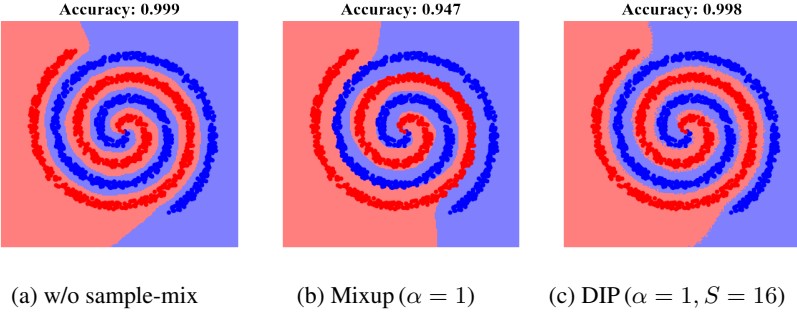

|  |  |  |
|:---:|:---:|:---:|
| **Accuracy: 0.999** | **Accuracy: 0.947** | **Accuracy: 0.998** |
| (a) w/o sample-mix | (b) Mixup ($\alpha = 1$) | (c) DIP ($\alpha = 1, S = 16$) |

Figure 1: Test accuracy and visualization of decision area on 2d spirals data. The neural networks are trained either (Left) without sample-mix, (Center) with the existing sample-mix method, or (Right) with the proposed sample-mix training + output approximation with monte carlo sampling. Although the standard sample-mix training deteriorates classification performance due to the biased decision boundary caused by sample-mix training, our method mitigates this problem. See section 2 for the details of a hyper-parameter of beta distribution $\alpha$ and the number of sampling $S$.

---

\*shima@ms.k.u-tokyo.ac.jp, {guguchi,hayasick,sosk}@preferred.jp

To overcome this problem, we propose a novel framework called Data Interpolating Prediction (DIP), where the sample-mixing process is encapsulated in a classifier. More specifically, we consider sample-mix as a stochastic perturbation in a function and obtain the prediction by computing the expected value over the random variable. Note that we apply sample-mix to both train and test samples in our framework. This procedure is similar to existing work such as monte carlo dropout (Gal & Ghahramani, 2016) and Augmented PAttern Classification (Sato et al., 2015). Furthermore, we establish the generalization error bound for our algorithm via Rademacher complexity and find that sample-mix helps to reduce the Rademacher complexity of a hypothesis class. Through experiments on benchmark image datasets, we confirm the generalization gap can be reduced by sample-mix and demonstrate the effectiveness of the proposed method.

## 2 PROPOSED METHOD

In this section, we propose a novel framework called Data Interpolating Prediction (DIP), where the sample-mixing process is encapsulated in the hypothesis class of a classifier. Furthermore, we theoretically establish the generalization bound for proposed method via Rademacher complexity and show that sample-mix is effective to reduce Rademacher complexity.

### 2.1 DATA INTERPOLATING PREDICTION

Let $\mathcal{X} \subset \mathbb{R}^d$ and $\mathcal{Y} \subset \{0,1\}^K$ be a $d$-dimensional input space and a $K$-dimensional one-hot label space. Denote a classifier by $f : \mathbb{R}^d \to \mathbb{R}^K$. The standard goal of classification problems is to obtain a classifier which minimizes the classification risk defined as follows:

$$L(f) := \mathbb{E}_{(\boldsymbol{x},\boldsymbol{y}) \sim p(\boldsymbol{x},\boldsymbol{y})}[\ell(f(\boldsymbol{x}), \boldsymbol{y})], \tag{1}$$

where $p(\boldsymbol{x}, \boldsymbol{y})$ is the joint density of an underlying distribution and $\ell$ is a loss function. Here, we consider a function where the sample-mixing process is encapsulated as a random variable. We describe sample-mix between $\boldsymbol{x}$ and $\boldsymbol{x}'$ with a function $\psi : \mathbb{R}^d \times \mathbb{R}^d \times \mathbb{R} \to \mathbb{R}^d$ as follows.

$$\psi(\boldsymbol{x}, \boldsymbol{x}', \lambda) := \lambda \boldsymbol{x} + (1 - \lambda)\boldsymbol{x}', \tag{2}$$

where $\lambda \in [0, 1]$ is a parameter that controls mixing ratio between two samples.

Let $(\boldsymbol{x}_1, \boldsymbol{y}_1), ..., (\boldsymbol{x}_n, \boldsymbol{y}_n)$ be i.i.d. samples drawn from $p(\boldsymbol{x}, \boldsymbol{y})$ , $\widehat{p}(\boldsymbol{x}) := \frac{1}{n} \sum_{i=1}^n \delta(\boldsymbol{x} = \boldsymbol{x}_i)$ be the density of the empirical distribution of $\{(\boldsymbol{x}_i, \boldsymbol{y}_i)\}_{i=1}^n$, and $h$ be a specified classifier. Using the function $\psi$, we define the deterministic function $f : \mathbb{R}^d \to \mathbb{R}^K$ by

$$f(\boldsymbol{x}) := \mathbb{E}_{\lambda \sim p(\lambda), \boldsymbol{x}' \sim \widehat{p}(\boldsymbol{x})} \left[ h(\psi(\boldsymbol{x}, \boldsymbol{x}', \lambda)) \right], \tag{3}$$

where $p(\lambda)$ is some density function over $[0, 1]$. Note that the function $f$ corresponds to the base function $h$ when we set $p(\lambda) = \delta(\lambda = 1)$.

### 2.2 PRACTICAL OPTIMIZATION

Since the expected value $\mathbb{E}_{\lambda \sim p(\lambda), \boldsymbol{x}' \sim p(\boldsymbol{x}')}[\cdot]$ is usually intractable, we train the classifier by minimizing upper the bound of an empirical version of $L(f)$,

$$\widehat{L}(f) := \frac{1}{n} \sum_{i=1}^n \ell(f(\boldsymbol{x}_i), \boldsymbol{y}_i) = \frac{1}{n} \sum_{i=1}^n \ell(\mathbb{E}_{\lambda \sim p(\lambda), \boldsymbol{x}' \sim \widehat{p}(\boldsymbol{x})} \left[ h(\psi(\boldsymbol{x}_i, \boldsymbol{x}', \lambda)) \right], \boldsymbol{y}_i). \tag{4}$$

By applying Jensen's inequality, we have

$$\widehat{L}(f) \leq \mathbb{E}_{\{\lambda_j\}_{j=1}^S \sim p(\lambda), \{\boldsymbol{x}'_j\}_{j=1}^S \sim \widehat{p}(\boldsymbol{x})} \left[ \frac{1}{n} \sum_{i=1}^n \ell \left( \frac{1}{S} \sum_{j=1}^S h(\psi(\boldsymbol{x}_i, \boldsymbol{x}'_j, \lambda_j)), \boldsymbol{y}_i \right) \right], \tag{5}$$

where $S$ is a positive integer which represents the number of sampling to estimate the expectation. We denote the RHS in equation 5 by $\widehat{L}_{\mathrm{upper},S}(f)$. The tightness of the above bound is related to the value of $S$ as

$$\widehat{L}(f) \leq \widehat{L}_{\mathrm{upper},S+1}(f) \leq \widehat{L}_{\mathrm{upper},S}(f). \tag{6}$$

We can prove this in a similar manner to Burda et al. (2016). Since $\lim_{S \to \infty} \widehat{L}_{\mathrm{upper},S}(f) = \widehat{L}(f)$, larger $S$ helps a precise risk estimation.

## 2.3 LABEL-MIXING OR LABEL-PRESERVING

There are two types of sample-mix data augmentation, namely, label-mixing approach and label-preserving approach. We can show that the objective functions of both approaches are consistent under some conditions.

**Proposition 1.** *Suppose that $\ell$ is a linear function with respect to the second argument and $p(\lambda) = \text{Beta}(\alpha, \alpha)$ for some constant $\alpha > 0$. Then we have the following equation.*

$$
\begin{aligned}
&\mathbb{E}_{\{(\boldsymbol{x},\boldsymbol{y}),(\boldsymbol{x}',\boldsymbol{y}')\}\sim p(\boldsymbol{x},\boldsymbol{y}),\lambda\sim\text{Beta}(\alpha,\alpha)}\left[\ell(h(\psi(\boldsymbol{x},\boldsymbol{x}',\lambda)),\lambda\boldsymbol{y}+(1-\lambda)\boldsymbol{y}')\right]\\
&= \mathbb{E}_{(\boldsymbol{x},\boldsymbol{y})\sim p(\boldsymbol{x},\boldsymbol{y}),\boldsymbol{x}'\sim p(\boldsymbol{x}),\lambda\sim\text{Beta}(\alpha+1,\alpha)}\left[\ell(h(\psi(\boldsymbol{x},\boldsymbol{x}',\lambda)),\boldsymbol{y})\right].
\end{aligned}
\tag{7}
$$

The proof of this theorem can be found in the blog post[1]. For many label-mixing approaches (Zhang et al., 2018; Verma et al., 2018), they use beta distribution for a prior of $\lambda$. Thus, the optimization of such approaches can be considered as a special case of our framework because an empirical version of RHS in equation 7 corresponds to $\widehat{L}_{\text{upper},1}$ where $p(\lambda)$ is set to $\text{Beta}(\alpha+1,\alpha)$. We experimentally investigate behaviors of both label-mixing and label-preserving training in Sec. 3.

## 2.4 GENERALIZATION BOUND VIA RADEMACHER COMPLEXITY

In this section, we present a generalization bound for a function equipped with sample-mix. Let $\mathcal{F}$ be a function class of the specified model and $\widehat{\mathfrak{R}}_n(\mathcal{F})$ be the empirical Rademacher complexity of $\mathcal{F}$. Then we have the following inequality.

**Proposition 2.** *Let $\{(\boldsymbol{x}_i,\boldsymbol{y}_i)\}_{i=1}^n$ be i.i.d. random variables drawn from an underlying distribution with the density $p(\boldsymbol{x},\boldsymbol{y})$ and $\ell \circ \mathcal{F} := \{\ell \circ f \mid f \in \mathcal{F}\}$. Suppose that $\ell$ is bounded by some constant $B > 0$. For any $\delta > 0$, with the probability at least $1 - \delta$, the following holds for all $f \in \mathcal{F}$.*

$$
\mathbb{E}_{(\boldsymbol{x},\boldsymbol{y})\sim p(\boldsymbol{x},\boldsymbol{y})}[\ell(f(\boldsymbol{x}),\boldsymbol{y})] \leq \frac{1}{n}\sum_{i=1}^n \ell(f(\boldsymbol{x}_i),y_i) + 2\widehat{\mathfrak{R}}_n(\ell \circ \mathcal{F}) + 3B\sqrt{\frac{\log\frac{2}{\delta}}{2n}}.
\tag{8}
$$

The proof of this theorem can be found in the textbook such as Mohri et al. (2012). Now we analyze a Rademacher complexity of a proposed function class. Let $\mathcal{H}$ be a specified function class and $\mathcal{F} := \{f(\boldsymbol{x}) = \mathbb{E}_{\lambda\sim p(\lambda),\boldsymbol{x}'\sim \widehat{p}(\boldsymbol{x})}[h(\psi(\boldsymbol{x},\boldsymbol{x}',\lambda))] \mid h \in \mathcal{H}\}$ as defined in equation 3. Suppose that the empirical Rademacher complexity of $\mathcal{H}$ can be bounded with some constant $C_\mathcal{H}$ as follows.

$$
\widehat{\mathfrak{R}}_n(\mathcal{H}) \leq \frac{C_\mathcal{H}}{n}\sqrt{\sum_{i=1}^n \|\boldsymbol{x}_i\|_2^2}.
\tag{9}
$$

We can prove this assumption holds for neural network models in a similar manner to Gao & Zhou (2016). Then we have the following theorem.

**Theorem 1.** *Suppose that $\ell$ is a $\rho$-Lipschitz function with respect to the first argument $(0 < \rho < \infty)$ and $\mathcal{H}$ satisfies the assumption in equation 9. Let $\mathcal{F} = \{f : \mathcal{X} \to \mathbb{R}\}$ be a function class of $f$ defined in equation 3. Then we have the following inequality.*

$$
\widehat{\mathfrak{R}}_n(\ell \circ \mathcal{F}) \leq \frac{\rho C_\mathcal{H}}{\sqrt{n}}\sqrt{C_\lambda \frac{1}{n}\sum_{i=1}^n \|\boldsymbol{x}_i\|_2^2 + (1-C_\lambda)\left\|\frac{1}{n}\sum_{i=1}^n \boldsymbol{x}_i\right\|_2^2},
\tag{10}
$$

*where $C_\lambda = \mathbb{E}_\lambda\left[\lambda^2 + (1-\lambda)^2\right]$.*

Note that $\frac{1}{n}\sum_{i=1}^n \|\boldsymbol{x}_i\|_2^2 \geq \|\frac{1}{n}\sum_{i=1}^n \boldsymbol{x}_i\|_2^2$ always holds from Jensen's inequality and $C_\lambda \in [0,1]$. Thus, sample-mix can reduce the empirical Rademacher complexity of the function class, which reduces the generalization gap (i.e., $L(f) - \widehat{L}(f)$). For example, when $p(\lambda) = \text{Beta}(\alpha+1,\alpha)$ in equation 3, we have $C_\lambda = \frac{\alpha+1}{2\alpha+1}$, which is a monotonically decreasing function with respect to $\alpha$. Hence, we claim that larger $\alpha$ can be effective for the smaller generalization gap. We experimentally analyze the behavior with respect to $\alpha$ in Sec. 3.

---

[1]inFERENCe, https://www.inference.vc/mixup-data-dependent-data-augmentation

## 3 EXPERIMENTS ON CIFAR DATASETS

In this section, we analyze the behavior of our proposed framework through experiments on CI-FAR10/100 datasets (Krizhevsky & Hinton, 2009). We evaluated the classification performances with two neural network architectures, VGG16 (Simonyan & Zisserman, 2015) and PreActRes-Net18 (He et al., 2016). The details of the experimental setting are described in Appendix B. For our proposed DIP, output after final fully-connected layer is used as $h(\boldsymbol{x})$ in equation 3 and the expected output is approximated by 500 times monte carlo sampling in test stage. As we discussed in Sec. 2.3, there are two types of optimization process when $S = 1$. We evaluated both label-preserving and label-mixing style training. We set $p(\lambda) = \text{Beta}(\alpha + 1, \alpha)$ for label-preserving sample-mix train-ing and $p(\lambda) = \text{Beta}(\alpha, \alpha)$ for label-mixing sample-mix training. Note that the prediction was computed with $p(\lambda) = \text{Beta}(\alpha + 1, \alpha)$ in test stage even when label-mix style was used for training.

For two baseline methods, we trained a classifier with (i) standard training (without sample-mix) and (ii) Mixup (Zhang et al., 2018) training (label-mixing style). To evaluate the performances of these methods, we computed the prediction only from clean samples. We used $p(\lambda) = \text{Beta}(\alpha, \alpha)$ for Mixup training.

We show the classification performances in Table 1 and generalization gap (i.e., the gap between train and test performances) in Fig. 2. Due to limited space, the magnified versions of experimental results are shown in Appendix C. As can be seen in Table 1, our proposed method is likely to outperform existing Mixup approach.

**Remarks:** For all approaches including existing Mixup, the larger $\alpha$ leads to the smaller generaliza-tion gap, which is consistent with the discussion in Sec. 2.4. In addition, we found that the larger $S$ is likely to enlarge the gap and deteriorate the performance on test samples. It might be because the variance of the empirical loss function computed by $S$ times sampling plays a role of regularization.

Table 1: Mean misclassification rate and standard error over three trials on CIFAR10/100 datasets.

| Model | Method | CIFAR10 | CIFAR100 |
|---|---|---|---|
| | without sample-mix | 6.78 (0.057) | 28.68 (0.169) |
| | Mixup ($\alpha = 1$) (Zhang et al., 2018) | 5.81 (0.031) | 26.58 (0.044) |
| VGG16 | DIP ($\alpha = 1, S = 1$, label-mixing) | 5.74 (0.100) | **25.48 (0.034)** |
| | DIP ($\alpha = 1, S = 1$, label-preserving) | 6.05 (0.015) | 26.57 (0.155) |
| | DIP ($\alpha = 2, S = 4$) | **5.52 (0.041)** | 26.73 (0.054) |
| | without sample-mix | 5.68 (0.015) | 25.25 (0.272) |
| | Mixup ($\alpha = 1$) | 4.46 (0.082) | 22.58 (0.074) |
| PreActResNet18 | DIP ($\alpha = 1, S = 1$, label-mixing) | **4.36 (0.079)** | **21.97 (0.052)** |
| | DIP ($\alpha = 1, S = 1$, label-preserving) | 4.83 (0.125) | 23.33 (0.052) |
| | DIP ($\alpha = 2, S = 4$) | 4.40 (0.036) | 22.04 (0.067) |

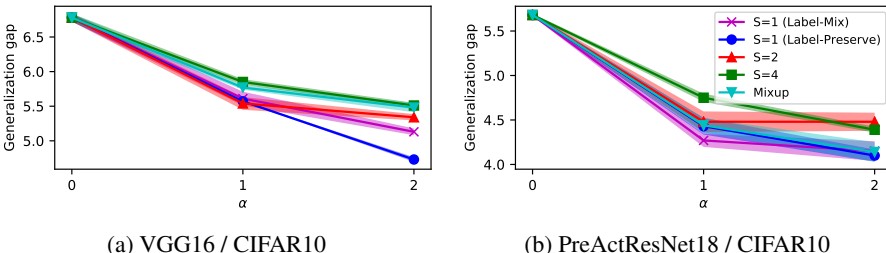

(a) VGG16 / CIFAR10                (b) PreActResNet18 / CIFAR10

Figure 2: Mean generalization gap and standard error over three trials on CIFAR10 dataset. $\alpha = 0$ indicates standard training without sample-mix.

## 4 CONCLUSION

In this paper, we proposed a novel framework called Data Interpolating Prediction (DIP), where sample-mix is encapsulated in the hypothesis class of a classifier. We theoretically established the generalization error bound via Rademacher complexity and showed that sample-mix is effective to reduce the generalization gap. Through experiments on CIFAR datasets, we demonstrated that our approach can outperform existing Mixup data augmentation.

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

## A PROOFS OF THEOREM 1

In this section, we give a complete proof of theorem 1. The empirical Rademacher complexity is defined as follows.

**Definition 1.** *Let $n$ be a positive integer, $\boldsymbol{x}_1, ..., \boldsymbol{x}_n$ be i.i.d. random variables drawn from $p(\boldsymbol{x})$, $\mathcal{F} = \{f : \mathcal{X} \to \mathbb{R}\}$ be a class of measurable functions, and $\boldsymbol{\epsilon} = (\epsilon_1, ..., \epsilon_n)$ be Rademacher random variables, namely, random variables taking $+1$ and $-1$ with the equal probabilities. Then the empirical Rademacher complexity of $\mathcal{F}$ is defined as*

$$\widehat{\mathfrak{R}}_n(\mathcal{F}) = \mathbb{E}_{\boldsymbol{\epsilon}} \left[ \frac{1}{n} \sup_{f \in \mathcal{F}} \sum_{i=1}^{n} \epsilon_i f(\boldsymbol{x}_i) \right]. \tag{11}$$

We assume that $\ell$ is a $\rho$-lipschitz function with respect to first argument. Here we have the following useful lemma. The proof of this lemma can be found in (Mohri et al., 2012).

**Lemma 1** (Talagrand's lemma). *Let $\Phi : \mathbb{R} \to \mathbb{R}$ be an $\rho$-lipschitz function. Then for any hypothesis set $\mathcal{H}$ of real valued function functions, the following inequality holds:*

$$\widehat{\mathfrak{R}}_n(\Phi \circ \mathcal{H}) \leq \rho \widehat{\mathfrak{R}}_n(\mathcal{H}). \tag{12}$$

From this lemma, we have

$$\widehat{\mathfrak{R}}_n(\ell \circ \mathcal{F}) \leq \rho \widehat{\mathfrak{R}}_n(\mathcal{F}). \tag{13}$$

Let $\mathcal{F} := \{f(\boldsymbol{x}) = \mathbb{E}_{\lambda \sim p(\lambda), \boldsymbol{x}' \sim \widehat{p}(\boldsymbol{x})} \left[ h(\psi(\boldsymbol{x}, \boldsymbol{x}', \lambda)) \right] \mid h \in \mathcal{H}\}$. In equation 9, we assume that

$$\widehat{\mathfrak{R}}_n(\mathcal{H}) = \mathbb{E}_{\boldsymbol{\epsilon}} \left[ \frac{1}{n} \sup_{h \in \mathcal{H}} \sum_{i=1}^{n} \epsilon_i h(\boldsymbol{x}_i) \right] \leq \frac{C_{\mathcal{H}}}{n} \sqrt{\sum_{i=1}^{n} \|\boldsymbol{x}_i\|_2^2}.$$

Now we can bound $\widehat{\mathfrak{R}}_n(\mathcal{F})$ as follows.

$$\widehat{\mathfrak{R}}_n(\mathcal{F}) = \mathbb{E}_{\boldsymbol{\epsilon}} \left[ \frac{1}{n} \sup_{f \in \mathcal{F}} \sum_{i=1}^{n} \epsilon_i f(\boldsymbol{x}_i) \right]$$

$$= \mathbb{E}_{\boldsymbol{\epsilon}} \left[ \frac{1}{n} \sup_{h \in \mathcal{H}} \sum_{i=1}^{n} \epsilon_i \mathbb{E}_{\lambda \sim p(\lambda), \boldsymbol{x}' \sim \widehat{p}(\boldsymbol{x})} \left[ h(\psi(\boldsymbol{x}_i, \boldsymbol{x}', \lambda)) \right] \right]$$

$$\leq \mathbb{E}_{\boldsymbol{\epsilon}, \lambda \sim p(\lambda), \boldsymbol{x}' \sim \widehat{p}(\boldsymbol{x})} \left[ \frac{1}{n} \sup_{h \in \mathcal{H}} \sum_{i=1}^{n} \epsilon_i h(\psi(\boldsymbol{x}_i, \boldsymbol{x}', \lambda)) \right] \quad (\because \text{convexity of sup})$$

$$\leq \frac{C_{\mathcal{H}}}{n} \mathbb{E}_{\lambda \sim p(\lambda), \boldsymbol{x}' \sim \widehat{p}(\boldsymbol{x})} \left[ \sqrt{\sum_{i=1}^{n} \|\psi(\boldsymbol{x}_i, \boldsymbol{x}', \lambda)\|_2^2} \right] \quad (\because \text{assumption in Eq. 9})$$

$$\leq \frac{C_{\mathcal{H}}}{n} \sqrt{\mathbb{E}_{\lambda \sim p(\lambda), \boldsymbol{x}' \sim \widehat{p}(\boldsymbol{x})} \left[ \sum_{i=1}^{n} \|\psi(\boldsymbol{x}_i, \boldsymbol{x}', \lambda)\|_2^2 \right]} \quad (\because \text{concavity of square root})$$

$$= \frac{C_{\mathcal{H}}}{n} \sqrt{\mathbb{E}_{\lambda \sim p(\lambda), \boldsymbol{x}' \sim \widehat{p}(\boldsymbol{x})} \left[ \sum_{i=1}^{n} \|\lambda \boldsymbol{x}_i + (1 - \lambda) \boldsymbol{x}'\|_2^2 \right]}$$

$$= \frac{C_{\mathcal{H}}}{n} \sqrt{\mathbb{E}_{\lambda \sim p(\lambda), \boldsymbol{x}' \sim \widehat{p}(\boldsymbol{x})} \left[ \sum_{i=1}^{n} \{\lambda^2 \|\boldsymbol{x}_i\|_2^2 + 2\lambda(1 - \lambda)\langle \boldsymbol{x}_i, \boldsymbol{x}' \rangle + (1 - \lambda)^2 \|\boldsymbol{x}'\|_2^2\} \right]}$$

$$= \frac{C_{\mathcal{H}}}{n} \sqrt{\mathbb{E}_{\lambda \sim p(\lambda)} \left[ \frac{1}{n} \sum_{i=1}^{n} \sum_{j=1}^{n} \{\lambda^2 \|\boldsymbol{x}_i\|_2^2 + 2\lambda(1 - \lambda)\langle \boldsymbol{x}_i, \boldsymbol{x}_j \rangle + (1 - \lambda)^2 \|\boldsymbol{x}_j\|_2^2\} \right]} \quad (\because \widehat{p}(\boldsymbol{x}) = \frac{1}{n} \sum_{i=1}^{n} \delta(\boldsymbol{x} = \boldsymbol{x}_i))$$

$$= \frac{C_{\mathcal{H}}}{\sqrt{n}} \sqrt{\mathbb{E}_{\lambda \sim p(\lambda)} \left[ \frac{\lambda^2}{n} \sum_{i=1}^{n} \|\boldsymbol{x}_i\|_2^2 + 2\lambda(1-\lambda)\langle \frac{1}{n} \sum_{i=1}^{n} \boldsymbol{x}_i, \frac{1}{n} \sum_{j=1}^{n} \boldsymbol{x}_j \rangle + \frac{(1-\lambda)^2}{n} \sum_{j=1}^{n} \|\boldsymbol{x}_j\|_2^2 \right]}$$

$$= \frac{C_{\mathcal{H}}}{\sqrt{n}} \sqrt{\mathbb{E}_{\lambda \sim p(\lambda)} [\lambda^2 + (1-\lambda)^2] \frac{1}{n} \sum_{i=1}^{n} \|\boldsymbol{x}_i\|_2^2 + E_{\lambda \sim p(\lambda)} [2\lambda(1-\lambda)] \langle \frac{1}{n} \sum_{i=1}^{n} \boldsymbol{x}_i, \frac{1}{n} \sum_{j=1}^{n} \boldsymbol{x}_j \rangle}$$

$$\leq \frac{C_{\mathcal{H}}}{\sqrt{n}} \sqrt{C_\lambda \frac{1}{n} \sum_{i=1}^{n} \|\boldsymbol{x}_i\|_2^2 + (1-C_\lambda) \left\| \frac{1}{n} \sum_{i=1}^{n} \boldsymbol{x}_i \right\|_2^2}.$$

By combining the above result and equation 13, we complete the proof of this theorem. $\qquad\square$

## B  DETAILS OF EXPERIMENTAL SETTING

In this section, we describe the details of training for experiments in Section 3.

### B.1  TRAINING

VGG16 (Simonyan & Zisserman, 2015) and PreActResNet18 (He et al., 2016) was used for experiments. We did not apply Dropout similarly to Mixup (Zhang et al., 2018). For all experiments, we trained a neural network for 200 epoch. Learning rate was set to $0.1$ in the beginning and multiplied by $0.1$ at 100 and 150 epoch. We applied standard augmentation such as cropping and flipping. The size of mini-batch was set to 128. We set $p(\lambda) = \text{Beta}(\alpha + 1, \alpha)$ for label-preserving sample-mix training and $p(\lambda) = \text{Beta}(\alpha, \alpha)$ for label-mixing sample-mix training. $\lambda$ was generated for each sample in mini-batch, and $\boldsymbol{x}'$ was obtained by permutation of samples in mini-batch.

### B.2  PREDICTION

For standard without sample-mix method and Mixup method, we predicted labels of test samples from clean samples. For proposed method, we predicted labels from the expectation over mixed samples computed by monte carlo approximation. In the same manner to the training process, we sampled $\lambda$ and $\boldsymbol{x}'$ 500 times and calculated the average to obtain the final output. In evaluation stage, data augmentation except for sample-mix was turned off.

## C    MAGNIFIED VERSIONS OF EXPERIMENTAL RESULTS

In this section, we present the magnfied version of experimental results in Section 3.

Table 2: Mean misclassification rate and standard error over 3 trials on CIFAR10/100 datasets.

| Model | Method | CIFAR10 | | CIFAR100 | |
|---|---|---|---|---|---|
| | | Train Acc. | Test Acc. | Train Acc. | Test Acc. |
| | without mix | 0.00 (0.000) | 6.78 (0.057) | 0.03 (0.003) | 28.68 (0.169) |
| | Mixup ($\alpha = 1.0$) | 0.05 (0.007) | 5.81 (0.031) | 0.27 (0.006) | 26.58 (0.044) |
| | Mixup ($\alpha = 2.0$) | 0.26 (0.029) | 5.73 (0.042) | 1.77 (0.108) | 26.34 (0.225) |
| | DIP ($\alpha = 1.0, S = 1$, label-mixing) | 0.13 (0.000) | 5.74 (0.100) | 0.48 (0.012) | 25.48 (0.034) |
| | DIP ($\alpha = 2.0, S = 1$, label-mixing) | 0.72 (0.035) | 5.85 (0.015) | 3.08 (0.147) | 25.45 (0.179) |
| VGG16 | DIP ($\alpha = 1.0, S = 1$, label-preserving) | 0.47 (0.012) | 6.05 (0.015) | 1.26 (0.072) | 26.57 (0.155) |
| | DIP ($\alpha = 2.0, S = 1$, label-preserving) | 2.08 (0.026) | 6.81 (0.046) | 7.38 (0.152) | 27.73 (0.140) |
| | DIP ($\alpha = 1.0, S = 2$) | 0.02 (0.003) | 5.57 (0.093) | 0.12 (0.006) | 25.87 (0.200) |
| | DIP ($\alpha = 2.0, S = 2$) | 0.30 (0.015) | 5.63 (0.032) | 1.15 (0.038) | 25.72 (0.042) |
| | DIP ($\alpha = 1.0, S = 4$) | 0.00 (0.000) | 5.85 (0.041) | 0.04 (0.003) | 27.20 (0.067) |
| | DIP ($\alpha = 2.0, S = 4$) | 0.01 (0.006) | 5.52 (0.041) | 0.10 (0.009) | 26.73 (0.054) |
| | without mix | 0.00 (0.000) | 5.68 (0.015) | 0.02 (0.000) | 25.25 (0.272) |
| | Mixup ($\alpha = 1.0$) | 0.02 (0.004) | 4.46 (0.082) | 0.09 (0.006) | 22.58 (0.074) |
| | Mixup ($\alpha = 2.0$) | 0.18 (0.013) | 4.32 (0.098) | 0.50 (0.027) | 22.87 (0.100) |
| | DIP ($\alpha = 1.0, S = 1$, label-mixing) | 0.09 (0.006) | 4.36 (0.079) | 0.26 (0.009) | 21.97 (0.052) |
| | DIP ($\alpha = 2.0, S = 1$, label-mixing) | 0.51 (0.013) | 4.66 (0.125) | 1.43 (0.029) | 22.31 (0.127) |
| PreActResNet18 | DIP ($\alpha = 1.0, S = 1$, label-preserving) | 0.40 (0.032) | 4.83 (0.125) | 0.78 (0.003) | 23.33 (0.052) |
| | DIP ($\alpha = 2.0, S = 1$, label-preserving) | 1.74 (0.022) | 5.84 (0.059) | 3.87 (0.023) | 23.75 (0.156) |
| | DIP ($\alpha = 1.0, S = 2$) | 0.02 (0.000) | 4.50 (0.116) | 0.09 (0.003) | 21.85 (0.231) |
| | DIP ($\alpha = 2.0, S = 2$) | 0.27 (0.010) | 4.75 (0.110) | 0.57 (0.003) | 21.94 (0.197) |
| | DIP ($\alpha = 1.0, S = 4$) | 0.00 (0.000) | 4.75 (0.046) | 0.03 (0.000) | 22.37 (0.136) |
| | DIP ($\alpha = 2.0, S = 4$) | 0.01 (0.003) | 4.40 (0.036) | 0.08 (0.007) | 22.04 (0.067) |

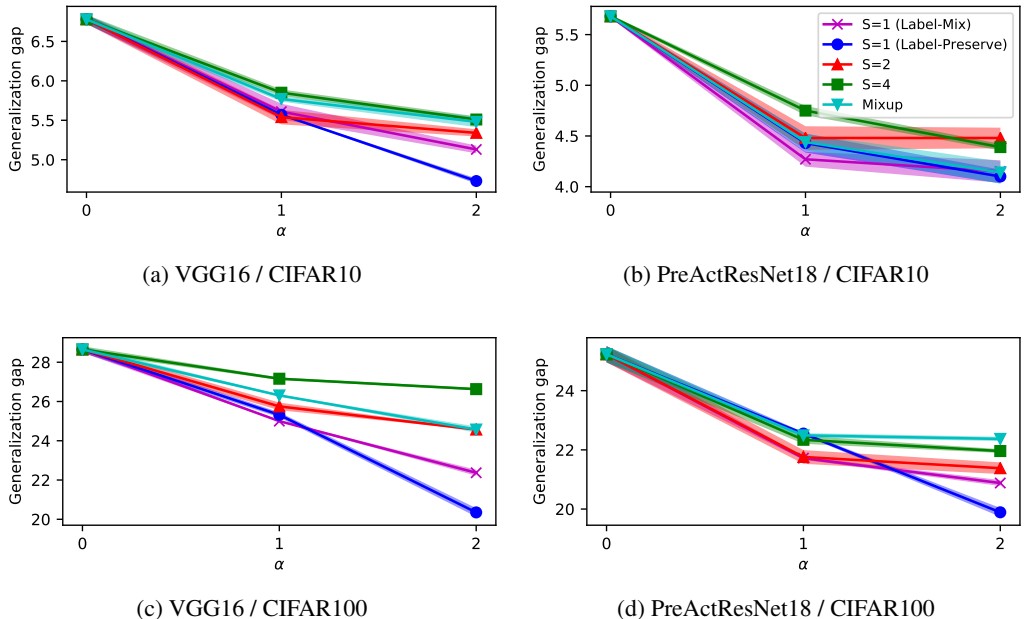

(a) VGG16 / CIFAR10

(b) PreActResNet18 / CIFAR10

(c) VGG16 / CIFAR100

(d) PreActResNet18 / CIFAR100

Figure 3: Mean generalization gap and standard error over three trials on CIFAR10/100 dataset. $\alpha = 0$ indicates standard training without sample-mix.

