# OpenReview forum: "Data Interpolating Prediction: Alternative Interpretation of Mixup"
_ICLR.cc/2019/Workshop/LLD — LLD 2019_

### Official Review · AnonReviewer2 · 2019-04-07
**A somewhat original approach, but insufficient experiments in order to draw conclusions regarding its value**

**Rating:** 3
**Confidence:** 2

**Review:**

The authors propose an alternative framework for applying the mixup method of [2]. To the best of my knowledge, this framework has not been propose before, hence the manuscript does offer some original ideas. That being said, the proposed framework is very related to established techniques, in particular to the practice of taking the average of the output of the classifier over a set of augmented version of a testing sample. The main practical contribution of the manuscript is the application of this technique: (a) for mixup and (b) at training time, too. On the downside, the reported results offer inconclusive evidence that this monte carlo sampling in training time can be a beneficial approach for classification. Generally, my main criticism is that I think that the experimental setup should have covered additional configuration combinations, so that the proposed framework is evaluated against suitable alternatives (details follow in the "Cons" section). In terms of clarity, I feel that the main ideas could have been presented briefly in a simpler language, before delving into the mathematical analysis.

Prons:
1. In addition to their experiments in CIFAR10 and CIFAR100, the authors briefly discuss the results on a simple synthetic dataset, which IMO illustrates nicely some potential dangers with mixup and it provides a rather clear motivation for this line of work.

2. Derivation of some theoretical results, which are interesting in their own right.

Cons:
IMO, there are quite a few methodological shortcomings:

1. The authors offer no justification of their choice of S=500 at testing time. I would expect a posteriori plot of how the testing accuracy changes for different values of S_testing given at least one training setting. As a related note when it comes to clarity, I assume that this sampling involves 500 cases from the training set, even at testing time. This is not made clear in the manuscript.

2. The main technique used by the paper (averaging the output of the classifier over a set of samples at both training and testing time) is rather orthogonal to mixup, to my understanding at least. In fact, the practice of generating augmented version of a given testing sample and taking the average of the output of the classifier over all the augmented versions is a well-known practice that has been shown to improve the testing performance in many practical scenarios (as an example, [1]). Therefore, I think that this method should have been applied to and compared with other augmentation methods, in addition to mixup. For example, what happens if the hypothesis classes is enriched with augmented versions of the same sample using more standard transformations (rotations, translations, etc) ?

3. It is my understanding that S=1 corresponds more or less to standard mixup, at training time at least. Thus, the difference of the configuration "Mixup (alpha=1)" and "DIP(alpha=1, S=1, label-mixing)" is only at testing time, with the sampling over the 500 mixup pairs that takes place in the second case. Since the experiments show that S=1 is the most performant configuration in most experiments, the value of the proposed monte carlo sampling at training time is not fully supported by the reported results, IMO. This should be discussed in the "Conclusion" section of the manuscript.

4. Dropout should have been used in the no-mixup baseline, as in the mixup paper [2].

5. I have the feeling that this work is only partially relevant to the scope of the workshop.

Nitpicking: I would like to mention a specific orthographic error that should be corrected: "monte carlo" is spelled "monte calro" for most of the manuscript.

[1] Jamaludin, Amir, Timor Kadir, and Andrew Zisserman. "SpineNet: Automated classification and evidence visualization in spinal MRIs." Medical image analysis 41 (2017): 63-73.

[2] Zhang, Hongyi, et al. "mixup: Beyond empirical risk minimization." arXiv preprint arXiv:1710.09412 (2017).

---

### Decision · Program_Chairs · 2019-04-16
**Acceptance Decision**

Accept